# PKC-Mediated Orai1 Channel Phosphorylation Modulates Ca^2+^ Signaling in HeLa Cells

**DOI:** 10.3390/cells11132037

**Published:** 2022-06-27

**Authors:** Ericka Martínez-Martínez, Víctor Hugo Sánchez-Vázquez, Daniel León-Aparicio, Jose Sanchez-Collado, Martín-Leonardo Gallegos-Gómez, Juan A. Rosado, Juan M. Arias, Agustin Guerrero-Hernández

**Affiliations:** 1Department of Biochemistry, Cinvestav, Mexico City 07360, Mexico; emartinezm@cinvestav.mx (E.M.-M.); vhsanchezv@cinvestav.mx (V.H.S.-V.); dalebx4@gmail.com (D.L.-A.); mlgallegos@cinvestav.mx (M.-L.G.-G.); 2Instituto de Física, Universidad Autónoma de San Luis Potosí, San Luis Potosí 78290, Mexico; 3Cellular Physiology Research Group, Department of Physiology, Institute of Molecular Pathology Biomarkers, University of Extremadura, 10003 Caceres, Spain; josesc@unex.es (J.S.-C.); jarosado@unex.es (J.A.R.); 4Programa de Neurociencias-UIICSE, Facultad de Estudios Superiores Iztacala, Universidad Nacional Autónoma de México, Tlalnepantla de Baz 54090, Mexico

**Keywords:** SOCE, phosphorylated Orai1, PKC, phosphomimetic

## Abstract

The overexpression of the Orai1 channel inhibits SOCE when using the Ca^2+^ readdition protocol. However, we found that HeLa cells overexpressing the Orai1 channel displayed enhanced Ca^2+^ entry and a limited ER depletion in response to the combination of ATP and thapsigargin (TG) in the presence of external Ca^2+^. As these effects require the combination of an agonist and TG, we decided to study whether the phosphorylation of Orai1 S27/S30 residues had any role using two different mutants: Orai1-S27/30A (O1-AA, phosphorylation-resistant) and Orai1-S27/30D (O1-DD, phosphomimetic). Both O1-wt and O1-AA supported enhanced Ca^2+^ entry, but this was not the case with O1-E106A (dead-pore mutant), O1-DD, and O1-AA-E106A, while O1-wt, O1-E106A, and O1-DD inhibited the ATP and TG-induced reduction of ER [Ca^2+^], suggesting that the phosphorylation of O1 S27/30 interferes with the IP_3_R activity. O1-wt and O1-DD displayed an increased interaction with IP_3_R in response to ATP and TG; however, the O1-AA channel decreased this interaction. The expression of mCherry-O1-AA increased the frequency of ATP-induced sinusoidal [Ca^2+^]_i_ oscillations, while mCherry-O1-wt and mCherry-O1-DD decreased this frequency. These data suggest that the combination of ATP and TG stimulates Ca^2+^ entry, and the phosphorylation of Orai1 S27/30 residues by PKC reduces IP_3_R-mediated Ca^2+^ release.

## 1. Introduction

Many different signals promote Ca^2+^ entry to the cell by activating diverse families of Ca^2+^ permeable channels. Some of them are the voltage-gated Ca^2+^ channels [1], the TRPC channels [2], the stretch channels [3], and the Orai channels [4,5]. The latter, together with the stromal interaction molecule (STIM), constitute what is known as store-operated Ca^2+^ entry or SOCE [6]. Accordingly, SOCE is a Ca^2+^ entry mechanism regulated by the filling state of the endoplasmic reticulum (ER) Ca^2+^ store [7]. The use of thapsigargin (TG), a potent and selective inhibitor of the sarco/endoplasmic reticulum Ca^2+^ ATPase (SERCA) pump, was instrumental in the characterization and identification of the molecular elements involved in SOCE [8,9].

The prevalent molecular mechanism for SOCE involves the participation of STIM1 as the Ca^2+^ sensor that couples the reduction in the free luminal Ca^2+^ concentration from the ER ([Ca^2+^]_ER_) with the activation of Orai channels at the plasma membrane [9,10,11]. STIM is a type 1 integral membrane protein with a single EF-hand domain facing the ER luminal side; when Ca^2+^ is bound to this domain, STIM stays in the dimeric form [12,13]. An [Ca^2+^]_ER_ reduction results in Ca^2+^ unbinding from the EF-hand and the aggregation of this protein, forming the puncta [14]. These aggregates result in the extension of the carboxy-terminal region of STIM1 to reach Orai1 channels at the plasma membrane. The interaction of the STIM1 Orai activating region (SOAR) or CRAC activation domain (CAD) with Orai1 results in this channel’s activation, which can be seen as a substantial but transient elevation of the cytoplasmic Ca^2+^ concentration ([Ca^2+^]_i_), remarkably when the SERCA pump is inhibited [15,16,17].

Interestingly, the overexpression of Orai1 results in a significant reduction of endogenous SOCE [18], i.e., Orai1 has a dominant-negative effect on SOCE, and the overexpression of both Orai1 and STIM1 is a requisite to obtain an increased SOCE activity [19]. The idea is that there is a dilution effect on STIM1 produced by the overexpression of the Orai1 channel. The excess of channels that modifies the STIM1-Orai1 stoichiometry implies fewer Orai1 channels with all their subunits bound to STIM1. This scenario might explain the negative effect on the SOCE of Orai1 overexpression [18,20].

The Ca^2+^ entering the cytoplasm by SOCE might then be captured by the SERCA pump to refill the ER Ca^2+^ store [21,22,23]. Because Orai1 channels appear to be near SERCA pumps [24], ER captures most of the Ca^2+^ entering through these channels, and consequently, Ca^2+^ does not reach the bulk of the cytoplasm [25]. This suggests an essential role for SOCE in refilling the ER Ca^2+^ store. In this sense, a decrease in the expression of STIM1 results in a sharp reduction in the agonist-induced Ca^2+^ release [26,27]. However, this reduction cannot be explained by a partly depleted ER Ca^2+^ store or reduced IP_3_-sensitivity of the IP_3_R [27]. Moreover, both isoforms of the STIM protein appear to have an inhibitory effect on agonist-induced Ca^2+^ release [28].

STIM1 and Orai1 form a complex that includes the IP_3_R, SERCA pump, TRP channels, and RACK—the receptor for activated C kinase [22,29,30,31]. Agonist-induced Ca^2+^ released from the ER involves IP_3_ production and the activation of IP_3_Rs. The ensuing increase in DAG and the elevation of [Ca^2+^]_i_ due to IP_3_ action activates PKC [32], and RACK should recruit this kinase to the complex mentioned above.

There are three different Orai channel isoforms, Orai1, 2, and 3 [4]. The Orai1 channel mRNA has two different translation initiation sites, and for this reason, cells express two Orai1 variants, the long one (α) and the short one (β). The β isoform is more prevalent than the α isoform [33,34]. The β isoform lacks the first 63 amino acids in the amino-terminal region, where the phosphorylation residues are located, although both isoforms interact with STIM1 equally well [33,35].

Regarding the effect of PKC on the Orai1 channel activity, this reflects only changes in Orai1α, as the Orai1β isoform lacks these phosphorylation sites. It has been shown that the inhibition of PKC-mediated phosphorylation of Ser27 and Ser30 residues of the Orai1α channel results in more extensive Ca^2+^ entry [36]. Furthermore, constitutively active PKC downregulates SOCE in different types of cancer cells [37]. However, Ser27 and Ser30 are the two main phosphorylation sites in the Orai1α isoform [36]. The nearby Ser34 can also be phosphorylated, but in this case, by PKA, which results in increased Ca^2+^-dependent inactivation [38]. This region rich in Ser residues has multiple roles. It interacts with adenylate cyclase [34,39] with the IP_3_R [29,30,31] and modulates channel inactivation [38].

We overexpressed three different Orai1 channels in HeLa cells to study their role in Ca^2+^ signaling, namely, the wild type Orai1 channel (Orai1-wt) and two different mutants of Ser27 and Ser30 residues, Orai1-AA (an Orai1 mutant that cannot be phosphorylated in these residues [36]) and a phosphomimetic mutant (Orai1-DD). To achieve the rapid and complete depletion of the ER Ca^2+^ store, we used the combination of ATP and TG to induce Ca^2+^ release [40]. Our data suggest that PKC-mediated Orai1 phosphorylation stimulates Ca^2+^ entry and reduces IP_3_R-mediated Ca^2+^ release. These data point to PKC as a significant regulator of Orai1 channel activity and indicate that the Orai1 channel regulates IP_3_R-mediated Ca^2+^ release by replenishing the ER Ca^2+^ store and modulating IP_3_R activity.

## 2. Results

### 2.1. The Orai1 Channel Overexpression Resulted in a Higher Agonist-Induced Ca^2+^ Entry and Reduced ER Ca^2+^ Depletion

The overexpression of O1-wt in HeLa cells did not modify the peak [Ca^2+^]_i_ response to the addition of either TG (1 μM) or ATP (10 μM) in the presence of 1.8 mM external [Ca^2+^] (Appendix A). To obtain a rapid and complete depletion of the ER Ca^2+^ store, we applied the combination of ATP and TG (Appendix A, black trace). The TG-induced reduction in the endoplasmic reticulum lumen [Ca^2+^] ([Ca^2+^]_ER_) was slower (Appendix A, blue trace) than the combination, while ATP alone produced a slower and incomplete reduction of [Ca^2+^]_ER_ (Appendix A, red trace). The application of ATP and TG increased the initial rate of the [Ca^2+^]_i_ response (Figure 1A) and inhibited the sustained reduction in [Ca^2+^]_ER_ seen with either Mag-fluo-4 (Figure 1C), a low-affinity Ca^2+^ indicator located in the ER [41], or with erGAP3 (Appendix A) [42], a ratiometric genetically encoded Ca^2+^ indicator targeted to the ER. These two effects were specific for the Orai1 channel, as the overexpression of Orai2 (Figure 1, O2) or Orai3 (Figure 1, O3) inhibited the rate of [Ca^2+^]_i_ rise, but they did not reduce the sustained [Ca^2+^]_ER_ decline (Figure 1). Remarkably, the overexpression of the dominant-negative, dead-pore mutant, Orai1 E106A (O1-E106A), inhibited the rate of [Ca^2+^]_i_ rise in response to ATP and TG. In contrast, [Ca^2+^]_ER_ reduction was still inhibited, similar to O1-wt, although O1-E106A forms a non-functional channel (Figure 1 pink traces and columns).

### 2.2. The Effects of the Orai1 Channel on Ca^2+^ Entry and the [Ca^2+^]_ER_ Required the Participation of Ca^2+^-Dependent PKCs

As ATP produces both IP_3_ and DAG, and the latter activates PKC, we decided to study the role of PKC both in the [Ca^2+^]_i_ and [Ca^2+^]_ER_ responses. The presence of Gö 6976, an inhibitor of Ca^2+^-dependent PKCs [43], inhibited Ca^2+^ entry because the rate of [Ca^2+^]_i_ rise was similar to the one seen with mock cells (Figure 2A,B, green traces and bars vs. black traces and bars). This PKC inhibitor also blocked the effect of Orai1 on the reduction of [Ca^2+^]_ER_ (Figure 2C,D, black and green traces and bars). Classical PKC activation requires an elevation of [Ca^2+^]_i_, and for this reason, we decided to test the effect of O1-wt overexpression in the absence of external [Ca^2+^]. In this case, the rate of [Ca^2+^]_i_ rise was significantly lower than the rate seen in the presence of external [Ca^2+^] and was not affected by the presence of O1-wt (Figure 2A,B, pink and brown traces and columns). The extent of [Ca^2+^]_ER_ reduction was larger in the absence than in the presence of external [Ca^2+^], although this larger reduction was not affected by the presence of O1-wt (Figure 2C,D, pink and brown traces and bars).

### 2.3. The Mutant Orai1 S27/30A Channel (O1-AA) Did Not Inhibit ER Ca^2+^ Release, While the Phosphomimetic S27/30D Channel (O1-DD) Inhibited ER Ca^2+^ Release without Any Participation of PKC

As PKC phosphorylates Orai1 on Ser27 and Ser30 amino acid residues, we decided to study whether this process is involved in reducing Ca^2+^ release and the higher Ca^2+^ transient by overexpressing O1-wt. The overexpression of the mutant O1-AA showed a similarly increased rate of [Ca^2+^]_i_ rise as for the O1-wt (Figure 3A,B, black traces and bars). This effect required a functional channel as the dominant-negative mutant O1-AA-E106A increased neither the peak nor the rate of [Ca^2+^]_i_ rise in response to ATP and TG (Appendix A). Moreover, the effect of O1-wt on the [Ca^2+^]_ER_ response was not present with the O1-AA mutant, as the [Ca^2+^]_ER_ reduction was similar to the one seen with mock cells (Figure 3C,D, black traces and bars). A completely different situation was observed with the overexpression of the O1-DD channel; both the rate of [Ca^2+^]_i_ rise, and the [Ca^2+^]_ER_ response were strongly reduced (Figure 3C,D, green traces and bars). The addition of Gö 6983, a broader PKC inhibitor than Gö 6976 [44], did not restore the inhibited ER Ca^2+^ release produced by the presence of the O1-DD channel (Figure 3C,D, pink traces and bars). The changes in the initial rate of [Ca^2+^]_i_ elevation correlated with those changes seen in the amplitude and time to peak of the ATP and TG induced [Ca^2+^]_i_ responses (Appendix A).

### 2.4. The Phosphorylation of the Orai1 Channel Enhanced Its Interaction with IP_3_Rs

As it has been shown that the application of agonists such as ATP induces the formation of a complex involving O1-wt, STIM, RACK, IP_3_Rs, and TRPC3 channels [29], we studied whether the Orai1 mutants used here, O1-AA and O1-DD, had any effect on the formation of this complex in response to the combination of ATP and TG. Initially, we corroborated the interaction between O1-wt and IP_3_R2 using immunoprecipitation [29]. The addition of ATP and TG increased the interaction between the endogenous Orai1 channel and IP_3_R2 (Appendix A, blue bar). In contrast, the overexpression of O1-wt itself increased this interaction (Appendix A, green bar) to the extent that now the addition of ATP and TG did not increase any further for this protein complex (Appendix A, white bar). The overexpression of the O1-DD channel did not show any apparent increase in complex formation in response to ATP and TG (Appendix A, blue bar). However, unexpectedly, the overexpression of the O1-AA channel resulted in a significantly decreased association with IP_3_R2 (Appendix A, white bar). Similarly, the interaction between O1-wt and IP_3_R1 showed a pattern resembling the one obtained for IP_3_R2. The overexpression of the O1-wt itself increased the formation of the Orai1 and IP_3_R1 complex compared with mock cells (Appendix A, green bar). The ATP and TG application strongly decreased the complex formation involving IP_3_R1 and the O1-AA mutant (Appendix A, white bar).

To further study the effect of ATP and TG on the interaction between Orai1 and IP_3_R2 (Figure 4), we carried out proximity ligation assays (PLA). The incubation for 3 min with ATP and TG resulted in a significantly increased number of dots per cell for a cell expressing either O1-wt (Figure 4A,B,G, red bars) or the O1-DD mutant (Figure 4C,D,G, blue bars). In contrast, it was the opposite situation for the O1-AA mutant (Figure 4E–G, black bars) as these agonists further decreased the number of dots/cell. Overall, these data indicate that the phosphorylation by PKC of Orai1 Ser27 and Ser30 residues stimulates the association with IP_3_Rs. Unexpectedly, the expression of the O1-AA channel decreased the complex formation in response to ATP and TG.

### 2.5. The O1-DD Channel Has a Higher Intracellular Localization Than O1-Wt and O1-AA Channels and a Lower SOCE

As Orai1 mutants showed different outcomes in their interaction with IP_3_Rs, we decided to study whether the Orai1 cellular distribution could explain the effect of the mutants in the complex formation. In agreement with previous observations [38,45], the overexpression of O1-wt fused to the mCherry protein at the N terminal end displayed a peripheral distribution (Figure 5A,B). It was also the case for the O1-AA mutant (Figure 5E,F). However, the O1-DD mutant showed a higher fraction of the channel in intracellular structures (Figure 5C,D). To determine whether these intracellular structures containing mCherry-O1-DD colocalizes with the ER, we conducted Manders’ studies between the Orai1 channel and ER tracker. All three channels, mCherry-O1-wt, mCherry-O1-AA, and mCherry-O1-DD, showed reduced colocalization with ER-tracker (Appendix A). As the O1-AA mutant supported SOCE [36], we tested whether the O1-DD mutant was functional in producing SOCE in response to TG alone. We corroborated that the overexpression of O1-wt alone, without STIM1, decreased SOCE (Appendix A), although the TG-induced reduction in the [Ca^2+^]_ER_ was the same for the O1-wt and mock cells (Appendix A). Thus, it was clear that we needed to combine the expression of Orai1 with STIM1, and we compared SOCE induced in mock cells (Figure 6A, red trace) with cells overexpressing only STIM1 (Figure 6A, blue trace), and the combination of STIM1 with either O1-wt (Figure 6A, black trace), O1-AA (Figure 6A, green trace), or O1-DD (Figure 6A, pink trace). In the absence of external Ca^2+^ and with 0.1 mM EGTA, the overexpression of Orai1 and STIM1 increased the resting [Ca^2+^]_i_, which was the O1-DD mutant that achieved the highest [Ca^2+^]_i_ in resting conditions (Figure 6B). However, the TG-induced [Ca^2+^]_i_ response in the absence of external [Ca^2+^] was the same in all cases (Figure 6C). The application of 2 mM external [Ca^2+^] to determine SOCE showed that O1-AA had the highest SOCE, followed by O1-wt, and O1-DD had the smallest SOCE (Figure 6D). These data indicate that even when O1-DD displayed a major intracellular distribution, it reached the plasma membrane at high enough levels to produce SOCE. Moreover, Western blot analysis showed that the O1-DD expression was significantly higher than the expression of the O1-AA channel (Appendix A). All these data indicate that the O1-DD channel is less effective at producing SOCE in comparison with the O1-wt or O1-AA channels.

### 2.6. O1-DD Channel Inhibited While O1-AA Stimulated the Frequency of ATP-Induced [Ca^2+^]_i_ Oscillations without TG

These data suggest that the phosphorylation state of the Orai1 channel interferes with agonist-induced Ca^2+^ signaling. To assess this effect, we loaded HeLa cells with Fluo-4, and we exposed them to 10 μM ATP to induce sinusoidal [Ca^2+^]_i_ oscillations, which were present in the majority of the mock cells (35 out of 39) with an averaged frequency of 4.4 oscillations in the 3-min evaluation period (Figure 7A, blue trace). The rest of the mock cells displayed a transient [Ca^2+^]_i_ elevation without oscillation (4 out of 39, Figure 7A,G red trace). The overexpression of O1-wt increased the number of cells with a single [Ca^2+^]_i_ response (15 out of 42 cells, Figure 7G). The rest of the cells expressing the O1-wt channel showed [Ca^2+^]_i_ oscillations with a similar frequency (3.4 oscillations in three min) as seen with the mock cells (Figure 7B,E). However, O1-wt changed the type of [Ca^2+^]_i_ oscillations from sinusoidal to baseline. Furthermore, the most frequent response was only two oscillations (Figure 7E, blue squares). Those cells overexpressing the O1-AA channel had the lowest number of oscillating cells (13 out of 28, Figure 7C,G). Still, those cells with [Ca^2+^]_i_ oscillations showed a sinusoidal form with the highest frequency (6.4 oscillations in four min). Moreover, the amplitude of the two types of [Ca^2+^]_i_ responses was enhanced by the expression of the O1-AA channel compared with those expressing the O1-DD channel (Figure 7C,F). The overexpression of the O1-DD channel resulted in the highest number of [Ca^2+^]_i_ oscillating cells (24 out of 30 cells, Figure 7D,G). Still, the O1-DD channel inhibited the frequency of [Ca^2+^]_i_ oscillations (2.4 oscillations in three min), and similarly to O1-wt, O1-DD changed the [Ca^2+^]_i_ oscillation to the baseline type. A single [Ca^2+^]_i_ oscillation was the most frequent event observed for those cells overexpressing the O1-DD channel (Figure 7E, black triangles).

## 3. Discussion

We studied the role of the phosphorylation of the Orai1 channel Ser27 and Ser30 residues in the agonist-induced Ca^2+^ responses in HeLa cells. We used the combination of ATP to activate purinergic receptors and thapsigargin (TG) to avoid SERCA pumps diverting Ca^2+^ entering the cytoplasm to the ER. We detected simultaneous changes in [Ca^2+^]_ER_ and [Ca^2+^]_i_ using the combination of Mag-Fluo-4 and Fura-2; the former readily enters the ER [41,46,47] to be used in intact cells as there is no need to permeabilize them, a condition needed when using Mag-Fura-2 [48]. However, the main caveat is that the ER does not retain Mag-Fluo-4, and the dye moves down to the secretory pathway [41]. Thus, erGAP3 [42], a ratiometric GECI targeted to the ER, showed that O1-wt overexpression led to a smaller ATP plus TG-induced [Ca^2+^]_ER_ reduction, as was the case with Mag-Fluo-4. Since only Mag-fluo-4 can be used together with Fura-2, we decided to use this combination for the rest of the experiments. ATP and TG combined in the mock-transfected cells resulted in a substantial [Ca^2+^]_ER_ depletion, as expected [4,41,49]. However, the new observation was that the presence of the O1-wt channel inhibited the ATP and TG-induced Ca^2+^ release. This inhibitory effect was specific to the O1 channel, as it was not observed with the other two Orai channel isoforms.

The inhibition by the O1-wt of the ATP plus TG-induced Ca^2+^ release required the presence of external Ca^2+^ and was blocked by a PKC inhibitor. Notably, a non-functional Orai1 channel, O1-E106A, also inhibited the agonist-induced Ca^2+^ release. Additionally, the O1-AA mutant—the channel that cannot be phosphorylated by PKC [36]—did not inhibit the agonist-induced ER Ca^2+^ release, while the phosphomimetic mutant (O1-DD) inhibited the ER depletion, even in the presence of a PKC inhibitor. These data suggest that agonist-induced, PKC-mediated phosphorylation of Ser27 and Ser30 inhibits agonist-induced IP_3_R-mediated Ca^2+^ release. Different agonists induce the interaction between Orai1 channels and IP_3_R [29,30,31]; in this work, the immunoprecipitation and proximity ligation assay showed that PKC-mediated phosphorylation of Ser27 and Ser30 residues is a critical step in promoting the interaction between Orai1 and IP_3_Rs. The interaction data showed that the O1 channel mutants were sensitive to ATP and TG. This result was unexpected, assuming that Orai1 phosphorylation was the only requisite for this interaction to occur. Therefore, these data mean that the interaction between Orai1 and IP_3_R includes additional factors. As STIM1 interacts with IP_3_R [26,27,28,29,30,50], it is quite feasible that the interaction between Orai1 and IP_3_R involves STIM1, among other elements.

It is now evident that the role of the Orai1 channel is richer than just refilling the ER Ca^2+^ store. It has been found in HEK cells that the absence of Orai1 reduces the frequency of agonist-induced Ca^2+^ oscillations [51]. However, this effect does not involve a lack of SOCE, as cells lacking both STIM1 and STIM2 showed a higher frequency of agonist-induced Ca^2+^ oscillations; this effect has not been explained because they could not observe a direct interaction between STIM and the IP_3_R channel [28]. Our data show that agonist-induced Orai1 phosphorylation stimulates the interaction of IP_3_R with this channel. We propose then that the long isoform of the Orai1 channel, which is the only one that can be phosphorylated by PKC [36], might be the link for the effect of STIM1 on the IP_3_R activity. These data suggest that the Orai1 α isoform has additional roles, allowing for the refilling of the ER Ca^2+^ store as it is involved in decreasing the IP_3_R-mediated Ca^2+^ release.

Many reports show that the Orai1 channel overexpression produces a potent inhibition of SOCE [18,52], a situation we confirmed in our experimental conditions. However, the combined application of ATP and TG resulted in a significantly faster and larger [Ca^2+^]_i_ response in cells overexpressing only functional Orai1 channels and in the presence of 1.8 mM CaCl_2_. Such an effect did not require the overexpression of STIM1, a requisite for Orai1 to increase SOCE [18]. The overexpression of either O2 or O3 did not replicate this elevated [Ca^2+^]_i_ response. This effect required a functional channel because of the absence of external [Ca^2+^] or overexpressing Orai1-E106A, a dominant-negative mutant of the Orai1 channel, resulting in a slower [Ca^2+^]_i_ response. This effect of the O1-E106A channel has two possible explanations; the inhibition of endogenous O1 channels or the inhibition of IP_3_R-mediated Ca^2+^ release. As the O1-AA-E106A mutant channel did not inhibit the agonist-induced [Ca^2+^]_i_ response, and O1-AA mutant had a decreased interaction with IP_3_R in response to the agonists, thus the data with O1-E106A support the idea that the O1 channel inhibits agonist-induced ER Ca^2+^ release. Furthermore, the expression of all three O1 channels resulted in a higher frequency of transient [Ca^2+^]_i_ response, particularly for the O1-AA channel. However, both O1-wt and O1-DD resulted in baseline [Ca^2+^]_i_ oscillating cells. These data suggest that the phosphorylated O1 channel stabilizes the inactivated form of IP_3_R, resulting in a decreased reduction in the luminal [Ca^2+^]_ER_, and this scenario might explain the effect of the agonist on the enhanced interaction between O1 and IP_3_R seen with PLA.

Our data suggest that this enhanced Ca^2+^ entry is seen with the overexpression of functional Orai1 channels involved in the phosphorylation of different amino acids beyond S27 and S30. The Orai1 mutant lacking the PKC-mediated phosphorylation sites, Ser27 and Ser30, displayed an enhanced Ca^2+^ entry, as seen with the O1-wt. This result agrees with previous publications where the O1-AA mutant supports SOCE equally well [36,37]. Interestingly, the inhibitor of PKC, Gö 6983, and the O1-DD mutant did not increase the agonist-induced Ca^2+^ entry. There is no clear explanation for this difference, unless the presence of both aspartates might become a steric hindrance for the phosphorylation of nearby Ser residues, for instance, Ser25. This is a hypothesis that needs to be tested in the future. It seems that the phosphorylation of these Ser residues stimulates agonist-induced Ca^2+^ entry by the Orai1 channel.

The interaction between STIM1 and the IP_3_Rs modulates the correct function of IP_3_Rs [26,27,28,53]. Downregulation of the STIM1 expression, but not STIM2, results in an inhibited agonist-induced [Ca^2+^]_i_ response [27]. This inhibition cannot be explained by reducing the IP_3_R sensitivity to its agonist or a reduction in the luminal [Ca^2+^]_ER_ [27]. Moreover, the agonist-induced Ca^2+^ release and the concomitant ER depletion are accelerated by the absence of both STIM isoforms [28]. Therefore, there is a fraction of STIM1 interacting with IP_3_Rs, and it is feasible that ATP and TG facilitate IP_3_R, STIM1, and Orai1 complex formation. This interacting protein complex results in a transiently increased Ca^2+^ entry. This scenario might explain why the combination of an agonist plus TG activates Ca^2+^ entry, a result not seen with TG alone. This result derives from the fact that TG does not activate IP_3_Rs [26,54]. However, it does not seem straightforward, as O1-DD interacts with IP_3_Rs better than O1-AA, yet O1-DD did not display the increased Ca^2+^ entry in response to ATP plus TG.

Ser27 and Ser30 residues are flanked by several positively charged amino acids, specifically arginine. It is feasible then that the sequence S27-R33 (S-R-R-S-R-R-R) from the Orai1 α isoform interacts with the plasma membrane PIP_2_ phospholipid [4,38]. In the case of O1-DD, this interaction might be weakened by the presence of two acidic residues. The interaction with PIP_2_ will be present for O1-wt and O1-AA channels. Therefore, it is important to study the role of these arginines in the O1 channel activity, particularly for the O1-DD mutant.

Cells express two isoforms of the Orai1 channel by alternative translation initiation, the long one (α isoform) susceptible to PKC-mediated phosphorylation and the short one (β isoform) that PKC cannot phosphorylate. Both isoforms are activated equally well by STIM1 [33,38], but PKC phosphorylates only the long variant [4]. We have shown that the phosphorylated channel interacts more with IP_3_R, reducing the IP_3_R-mediated Ca^2+^ release. This inhibitory effect of the phosphorylated O1 channel should facilitate the ER Ca^2+^ store refilling. This scenario suggests that the α isoform helps refill the ER Ca^2+^ store by allowing Ca^2+^ entry at the plasma membrane, i.e., SOCE, and inhibiting the Ca^2+^ releasing activity of IP_3_Rs. Our working hypothesis is that phosphorylated O1 stabilizes the Ca^2+^-induced inactivated state of IP_3_Rs. This scenario might be the reason for cells expressing both Orai1 channel isoforms.

## 4. Material and Methods

### 4.1. Materials

The BCA protein assay kit, Dulbecco’s Modified Eagle Medium, Penicillin Streptomycin antibiotic, Fura-2 AM, Mag-Fluo-4 AM, and SuperSignal West Dura chemiluminescent substrate were purchased from Thermo Fisher Scientific (Waltham, MA, USA). Lipofectamine 2000 and mouse anti-Flag antibody (MA1-91878) were bought from Invitrogen. Actin antibody (catalog A2066 of human β-actin) was purchased from Sigma (Madrid, Spain). Duolink Proximity Ligation Assay, ATP, PBS, and rabbit anti-Orai1 antibody (O8264) were acquired from SIGMA ALDRICH. Rabbit anti-IP_3_R1 (A302-157A) was from Bethyl Laboratories and mouse anti-IP_3_R2 (sc-39843) antibody was purchased from Santa Cruz Biotechnology. HeLa cells were obtained from ATCC. HeLa cells were transfected with eYFP-O1 where indicated, as previously described [55].

### 4.2. Cell Culture and Transfection

HeLa cells were cultured in 60 mm Petri dishes in Dulbecco’s Modified Eagle Medium-high glucose (DMEM, 25 mM) supplemented with 10% fetal bovine serum, 50 U/mL penicillin, and 50 μg/mL streptomycin and incubated at 37 °C in an atmosphere of 5% CO_2_. When the cells reached 70% confluence, the cell monolayer was transfected with Opti-MEM reduced serum media and a 1:5 μg/μL mix of the plasmid/Lipofectamine 2000™ for 24 h.

### 4.3. [Ca^2+^]_i_ and [Ca^2+^]_ER_ Determination in Cell Populations by Spectrofluorometry

After transfection, HeLa cell monolayers were washed twice with PBS, followed by trypsinization for 1.5 min. The cell suspension was placed in DMEM supplemented serum and microfuged at 400× *g* for 3.5 min. Then, the cell pellet was resuspended in a saline solution containing 121 mM NaCl, 5.4 mM KCl, 0.8 mM MgCl_2_, 6 mM NaHCO_3_, 1.8 mM CaCl_2_, 5.5 mM glucose, and 25 mM Hepes (pH 7.3 adjusted with NaOH). The Ca^2+^-free saline solution is the same without the CaCl_2_ addition and supplemented with 0.1 mM EGTA. Cell viability was always checked using the trypan blue exclusion assay. After that, 1 × 10^6^ cells/mL were incubated with 1 μM Fura-2 AM and 1 μM Mag-Fluo-4 AM in the dark at room temperature for 2 h. Most of Fura-2 remained in the cytoplasm, while most of Mag-Fluo-4 was located in the endoplasmic reticulum in our loading conditions [41,46,47,56]. Time course fluorescence recordings were done as previously described [54]. Briefly, HeLa cells (2 × 10^5^ cells/mL) resuspended in the same saline solution described above were placed in a round cuvette in a fluorescence spectrophotometer (PTI) with magnetic stirring and running at a 2.67 Hz. Fura-2 was excited at 340 nm, 360 nm, and 380 nm, while Mag-Fluo-4 was excited at 495 nm; the fluorescence signal was collected at 510 nm for both fluorophores. The fluorescence signal was allowed to stabilize for 10 min before cell stimulation. The values obtained during this period represent the resting [Ca^2+^]_i_ and the signal was used to normalize the changes in the Mag-Fluo-4 fluorescence. The Fura-2 ratio (340/380) was transformed into [Ca^2+^]_i_ using Grynkewicz’s equation [56], [Ca^2+^]_i_ = K_d_β(R − R_min_)/(R_max_ − R), and the FeliX32 analysis software (PTI), where R_max_ is the fluorescence ratio after the addition of digitonin (200 μM) and the R_min_ is the fluorescence ratio after the addition of EGTA (5 mM). This calibration was carried out at the end and for all of the experiments. These ratios were adjusted for viscosity using the correction factor β = 0.75 [57] and a K_d_ for Fura-2 of 200 nM. The background fluorescence was obtained using 12 mM MnCl_2_; these values were subtracted from 340 and 380 signals before calculating the ratios. The fluorescence signal reported by Mag-Fluo-4 was normalized to the basal time course (F_0_) employing a nonlinear fitting algorithm, and it was expressed as a fraction (∆F/F_0_).

### 4.4. Point Mutations of Orai1 Channel

We introduced two points mutations (S27D/S30D) in the plasmid p3XFlag-CMV-10-Orai1-S27A/S30A—A kind gift by Dr. Stefan Feske [36]—and the mutation E106A on the plasmid p3XFlag-Orai1-S27A/S30A to obtain the Orai1-S27A/S30A/E106A. To carry out S27D/S30D point mutations, we employed the PCR-driven overlapping extension assay [58] with the sense primer GCGGCGACCGCCGGGACCGCCG and the antisense primer CGGCGGTCCCGGCGGTCGCCGC. The fragment with the mutations was amplified by PCR and purified with the Kit GeneJET PCR Purification (Thermo Fisher) according to the manufacturer’s instructions; after digestion with HindIII, a 273 bp fragment was ligated to the complement of the channel’s cDNA. Mutation E106A was generated with the sense primer GGTGGCAATGGTGGCCGTGCAGCTGGACGC and the antisense primer GCGTCCAGCTGCACGGCCACCATTGCCACC; the mutated fragment was amplified by PCR and purified with the Kit GeneJET PCR Purification (Thermo Fisher). The construction 3XFlag-Orai1-S27A/S30A-E106A was carried out by ligating the mutated fragment (931 bp) with SacI sites at both ends.

The constructions for mCherry-Orai1-S27A/S30A and mCherry-Orai1-S27D/S30D were carried out using p3XFlag-CMV-7.1-mCherry-Orai1. In this case, we replaced the Orai1-wt cDNA with those for the mutations (either S27A/S30A or S27D/S30D) employing the unique BspEI (codifying region) and BamHI (polylinker) sites. Sanger’s DNA sequencing was used to verify the mutations and for correct ligation for all the new constructions.

CFP-Orai1, p3XFlag-CMV-7.1-mCherry-Orai1, and STIM1-CFP were kind gifts from Luis Vaca (Institute of Cellular Physiology, Mexico City, Mexico).

### 4.5. Immunoprecipitation and Western Blot for Orai1 Channel

Immunoprecipitation and Western blot were carried out as previously described [29,59]. Briefly, cells were transfected and grown on flasks until they reached 90% confluence; the trypsinized monolayer was resuspended in HEPES-buffered saline (125 mM NaCl, 5 mM KCl, 1 mM MgCl_2_, 5 mM glucose, 25 mM Hepes and 1.8 mM CaCl_2_, pH 7.4 adjusted with NaOH) and kept for 10 min at 37 °C followed by ATP (10 μM) plus TG (1 μM) stimulation for 3 min. Immediately after stimulation, the cells were lysed with a mix of 2× NP-40 lysis buffer, EDTA-free proteases inhibitors, and 2 mM Na₃VO₄ on an ice bed. The samples were sonicated and centrifuged (1.6 × 10^4^× *g* for 30 min at 4 °C). The protein pellets were quantified with the BCA protein assay kit and immunoprecipitated with 1 μg of either anti-Orai1 or anti-Flag antibodies to distinguish the Orai1 mutants that have a Flag tag at the amino-terminal domain and 30 μL of protein A agarose by overnight incubation at 4 °C in an oscillating rotor. The microfuged beads (16,000× *g* at 4 °C) were washed twice with cold PBS, and the proteins were denatured with Laemmli’s sample buffer supplemented with 5% DTT and 5% β-mercaptoethanol at 70 °C for 10 min followed by placing them on ice for 10 min. The supernatant of microfuged beads (10,000× *g*) was separated in SDS-PAGE (10% acrylamide gel) followed by blotting to a nitrocellulose membrane in semi-humid conditions (0.8 mA/cm^2^ of blotting paper). Membranes were blocked by incubating them with 10% (*w*/*v*) BSA diluted in TBST (Tris-buffered saline with 0.1% Tween-20) solution for 1 h at room temperature. Next, they were incubated with primary antibody diluted (1:500) in 10% BSA overnight at 4 °C, followed by 6× washes with TBST and then incubated for 1 h with horseradish peroxidase-conjugated goat anti-mouse IgG antibody (1:10,000 in TBST), followed by 6× washes and incubated with SuperSignal West Dura chemiluminescent substrate for 5 min.

Bands were quantified with a C-DiGit Chemiluminescent Western Blot Scanner (LI-COR Biosciences, Lincoln, NE, USA). The data shown are the ratio of IP_3_R signal/Orai1 signal.

### 4.6. Proximity Ligation Assay to Assess the Association between Orai1 and IP_3_R

Proximity Ligation Assay was adapted from the manufacturer’s instructions, as previously described [59], with minor modifications. Briefly, cells were grown on a coverslip to 80% confluence, followed by ATP (10 μM) plus TG (1 μM) stimulation for 3 min. Then, cells were fixed with 4% paraformaldehyde solution for 10 min and washed 2× with PBS for 5 min. Next, the cells were permeabilized with 0.5% Triton X-100 in PBS solution for 10 min and washed 3× with Tris-buffered saline (TBST, 0.05 % Tween 20) at room temperature. The coverslip was covered with a Duolink blocking solution and incubated in a humidified chamber for 60 min at 37 °C. The excess blocking solution was removed before adding the antibodies. These were rabbit anti-Orai1 and mouse anti-IP_3_R2, diluted 1:50 in a Duolink antibody solution and incubated for 60 min at 37 °C in a humidified chamber. Next, the coverslip was washed twice with Wash Buffer A (5 min at room temperature) and incubated with PLA probes (PLUS and MINUS PLA probes) for 60 min at 37 °C in the humidified chamber. The sample was washed twice with Wash Buffer A (5 min at room temperature) and incubated with ligase diluted in 1× Duolink ligation buffer at 1:40 for 60 min at 37 °C, followed by 2× washes with Wash Buffer A (5 min at room temperature). The amplification reaction was carried out with diluted polymerase (1:80 with 1× amplification buffer) in the dark at 37 °C for 100 min. Next, the sample was washed with Wash Buffer B for 10 min, and one additional wash was done with 0.01× Wash Buffer B for 1 min at room temperature. The excess of Wash Buffer B was removed and followed by Duolink in situ Mounting Medium with DAPI. The samples were analyzed in an epifluorescence inverted microscope (Nikon Eclipse Ti2, Amsterdam, The Netherlands) with an image acquisition and analysis system for video microscopy (NIS-Elements Imaging Software, Nikon). The PLA signal was excited at 594 nm and collected at 624 nm wavelengths, in addition to DAPI signals and brightfield images of cells.

### 4.7. Confocal Imaging of Transfected HeLa Cells

Previously, HeLa cells were grown on 21 mm × 21 mm coverslips, and once they reached 50% confluency, were transfected with 2 μg of DNA of either an empty vector (mock), mCherry-Orai1 wt, mCherry-Orai1-S27D/S30D, or mCherry-Orai1-S27A/S30A for 24 h with Lipofectamine 2000, as described above. HeLa cells were fixed with 2 mL per well of 4% (PBS) cold paraformaldehyde for 10 min at room temperature, follow by two PBS washes and permeabilization with 2 mL per well of 0.5% (PBS) Triton X-100 for 10 min at room temperature. The cells were washed three times with PBS and incubated with DAPI solution (DUO82040, Sigma) for 15 min. HeLa cell imaging was carried out with a ZEISS LSM700 confocal microscope using 63× objective (NA 1.4). The fluorescence signal from the mCherry protein was excited with the 555 nm laser line at one Airy unit (45 μm) and collected at 580 nm. The image dimensions were 1024 × 1024 pixels, 40 nm pixel size, and 12-bit resolution; these images were then deconvolved (Parallel Iterative Deconvolution 2D plugin by ImageJ) and their intensity profiles were obtained to visualize the Orai1 channel distribution for different mutants. All the data were analyzed with ImageJ software and were scaled in arbitrary units of fluorescence.

Confocal calcium imaging. HeLa cells grown and transfected as indicated above were loaded with 1 μM Fluo-4/AM at room temperature for 30 min in the dark. After this time, the cells were washed once with a recording saline solution with Ca^2+^. Images were acquired with one track, two channels, ex/em at 488/510 nm for Fluo-4 and 555/580 nm for mCherry signals. Image sizes were 512 × 512 pixels with a pixel size of 200 nm and 8 bit resolution. A cytoplasmic mask was used to calculate ∆(F − F_min_)/(F_0_ − F_min_) at a 5-s resolution time during the 4 min recording time. The F_min_ was the fluorescence signal in the absence of a cell. The average fluorescence for 1 min before the addition of 10 μM ATP was considered Fo.

### 4.8. Data Statistical Analysis

The data shown are the mean ± SEM, with n being the number of independent experiments carried out for each condition. Depending on the experimental design, data were analyzed with either unpaired *t*-tests with one-tail or one-way ANOVA with Dunnett’s post hoc test (GraphPad Prism 5.0). The difference was statistically significant if the null hypothesis had a * *p* < 0.05.

## Figures and Tables

**Figure 1 cells-11-02037-f001:**
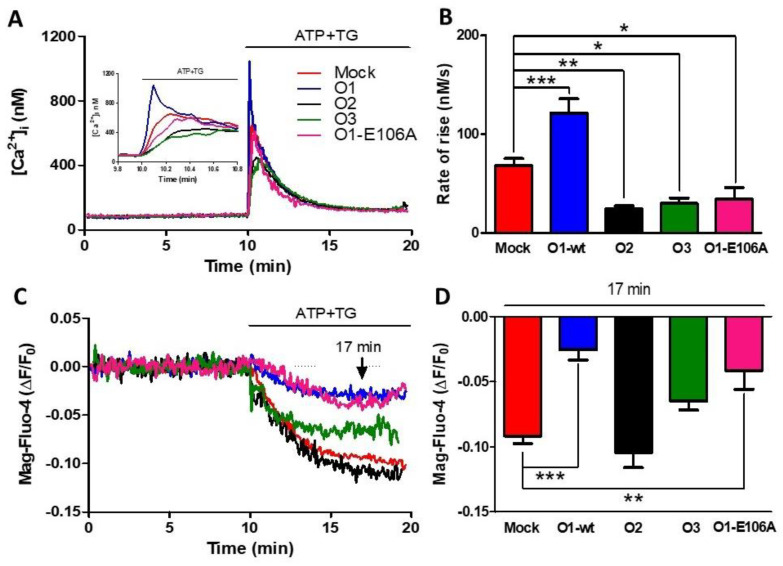
The dominant-negative mutant of the Orai1 channel (O1-E106A) inhibits the rate of agonist-induced Ca^2+^ release. (**A**) Time course of changes in [Ca^2+^]_i_ (assessed with Fura-2) in response to the 10 µM ATP and 1 μM TG application, where indicated, in a 1.8 mM Ca^2+^ medium for either mock cells (red trace) or cells overexpressing Orai1 (O1-wt, blue trace), Orai2 (O2, black trace), and Orai3 (O3, green trace) channels, in addition to cells overexpressing the dominant-negative mutant Orai1 E106A (O1-E106A, pink trace). The inset shows the initial ATP and TG induced [Ca^2+^]_i_ response time course. (**B**) The bars indicate that the average initial rate of [Ca^2+^]_i_ rise for the Orai1 channel (blue) was significantly higher than the mock cells (red), while O2 (black) and O3 overexpressing cells (green) had a significantly slower increase in [Ca^2+^]_i_. The rate of [Ca^2+^]_i_ rise for O1-E106A (pink) was significantly lower than for the mock cells. Data are presented as mean ± SEM. (**C**). The time course of Mag-Fluo-4 fluorescence reduction in response to the application of ATP-TG depicts changes in the [Ca^2+^]_ER_ in the same cells reporting the changes in the [Ca^2+^]_i_ (panel A). (**D**) Bar graph showing the average reduction in the Mag-Fluo-4 fluorescence sampled 7 min after the addition of ATP-TG (arrow). The overexpression of both O1-wt and O1-E106A resulted in a significantly smaller rate of decline in [Ca^2+^]_ER_. This reduction was not present for the O2 and O3 channels. Statistical analysis was carried out with the one-way ANOVA test and Dunnett’s correction for multiple comparisons. * *p* < 0.05; ** *p* < 0.01; *** *p* < 0.001. The number of independent experiments was *n* = 10 for mock cells and 5 for the rest. The bars represent the mean ± SEM.

**Figure 2 cells-11-02037-f002:**
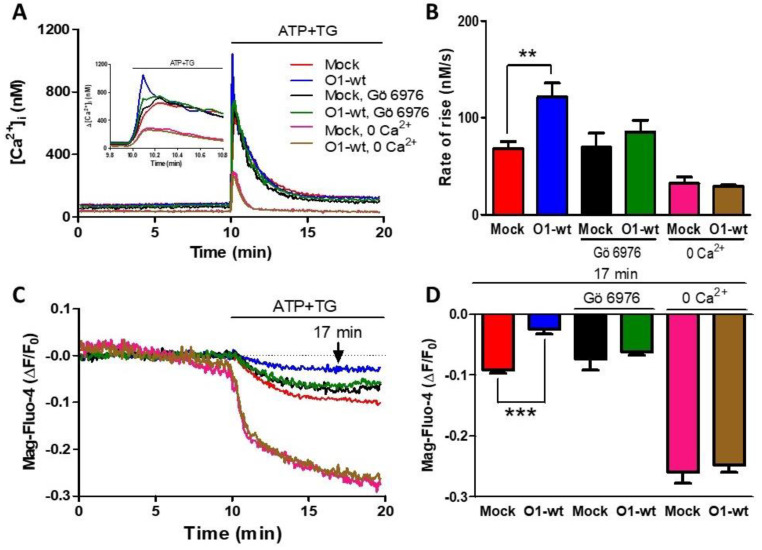
The inhibition of ER Ca^2+^ depletion by the Orai1 channel requires PKC activation and the presence of external [Ca^2+^]. (**A**) Time course of the [Ca^2+^]_i_ response to the addition of ATP-TG for either untreated mock cells (red trace) or mock cells in the presence of 200 nM Gö 6976 (black trace) or the absence of external Ca^2+^ (pink trace). The same conditions were applied to cells overexpressing the O1-wt channel: no treatment (blue trace); 200 nM Gö 6976 (green trace) or in the absence of external Ca^2+^ (brown trace). The inset shows the time course of the initial increase in the [Ca^2+^]_i_ triggered by the addition of ATP and TG. (**B**) Bar graphs show the average initial rate of [Ca^2+^]_i_ rise for those conditions displayed in panel A. Data are the mean ± SEM. The student’s *t*-test for one tail was used to compare each treatment with its corresponding mock. Both conditions inhibited the increase in the rate of [Ca^2+^]_i_ rise seen with O1-wt. (**C**) The time course of the reduction in the [Ca^2+^]_ER_ induced by ATP and TG for those cells depicted in panel A. (**D**) Bar graphs showing the Mag-fluo-4 fluorescence at 7 min after the addition of ATP-TG. ** *p* < 0.01; *** *p* < 0.001. The number of independent experiments was *n* = 10 for mock cells and 5 for the rest. Bars represent the mean ± SEM.

**Figure 3 cells-11-02037-f003:**
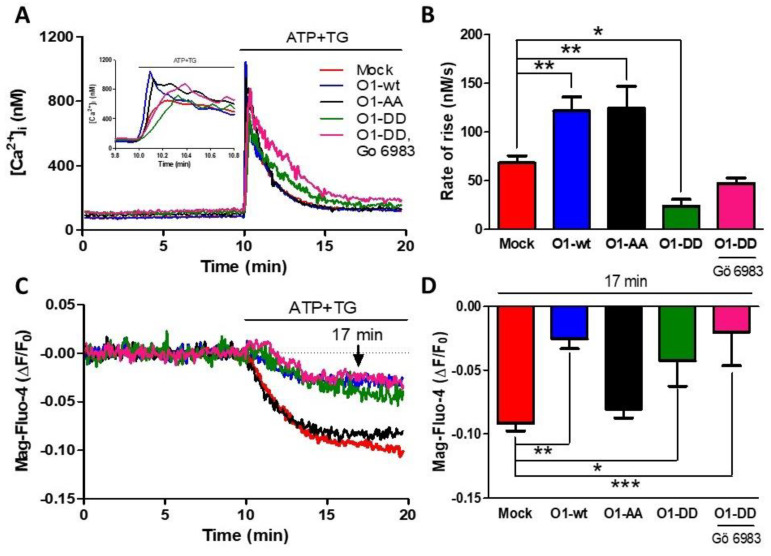
The S27/30A O1 mutant (O1-AA) increases while the S27/30D O1 mutant (O1-DD) inhibits the ATP and TG-induced rate of Ca^2+^ release. (**A**) Time course of the changes in [Ca^2+^]_i_ induced by the addition of ATP-TG for either mock cells (red trace) or cells overexpressing the wild type Orai1 channel (O1-wt, blue trace), double mutant Orai1 S27A/S30A (O1-AA, black trace), and phosphomimetic mutant Orai1-S27D/S30D alone (O1-DD, green trace) or together with 200 nM Gö 6983 (pink trace). The inset shows the initial increase in [Ca^2+^]_i_ resulting from the addition of ATP and TG. (**B**) The average initial rate of [Ca^2+^]_i_ rise for these cells is shown in panel A. Data are the mean ± SEM. (**C**) The time course of the [Ca^2+^]_ER_ reduction in response to ATP-TG application for the same cells shown in panel A. (**D**) Average Mag-fluo-4 ∆F/F_0_ decrease after 7 min of ATP-TG application. Statistical analysis was carried out with one-way ANOVA with Dunnett’s correction for multiple comparisons with the mock response. Bars represent the mean ± SEM. * *p* < 0.05; ** *p* < 0.01; *** *p* < 0.001. The number of independent experiments was *n* = 10 for mock cells, 4 for O1-DD, and 5 for the rest.

**Figure 4 cells-11-02037-f004:**
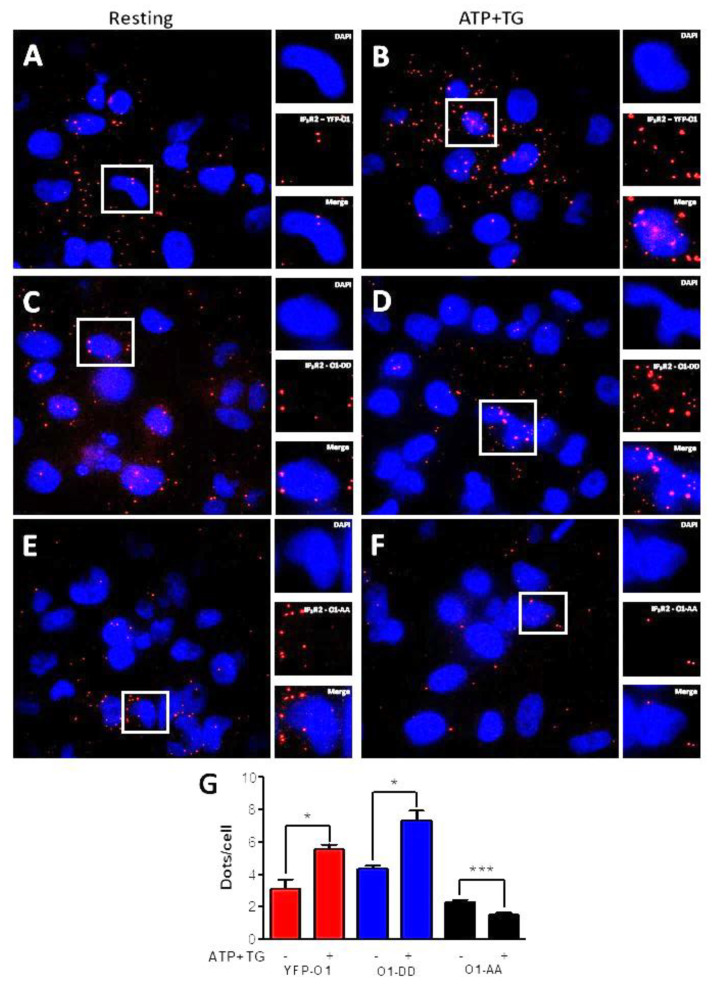
ATP and TG stimulate the interaction of IP_3_R2 with O1-wt and O1-DD, but inhibit the interaction of IP_3_R2 with O1-AA. A proximity ligation assay (PLA) was carried out in cells expressing either eYFP-Orai1, Orai1-S27D/S30D (O1-DD), or Orai1-S27A/S30A (O1-AA) for detecting their interaction with IP_3_R2. (**A**) Fluorescence images show nuclei stained with DAPI, and red dots represent positive PLA for IP_3_R2 and Orai1 for cells expressing eYFP-O1 at resting conditions or (**B**) after being stimulated with ATP (10 µM) and TG (1 µM) for 3 min. (**C**,**D**) Dot PLA images for cells expressing O1-DD channels at rest and after being stimulated with ATP and TG, respectively. (**E**,**F**) PLA images for cells expressing O1-AA channels at rest and after stimulation with ATP and TG, respectively. (**G**) The average number of red dots per nucleus as an indicator of the IP_3_R2–Orai1 interaction for cells overexpressing wild type Orai1 (eYFP-O1, red bars), phosphomimetic Orai1 (O1-DD, blue bars), and phosphorylation-resistant Orai1 mutant (O1-AA, black bars). Statistical analysis was carried out with an unpaired, one-tail Student’s *t*-test with the corresponding control. Bars show the mean ± SEM. * *p* < 0.05; *** *p* < 0.001 from at least 50 cells from three independent experiments.

**Figure 5 cells-11-02037-f005:**
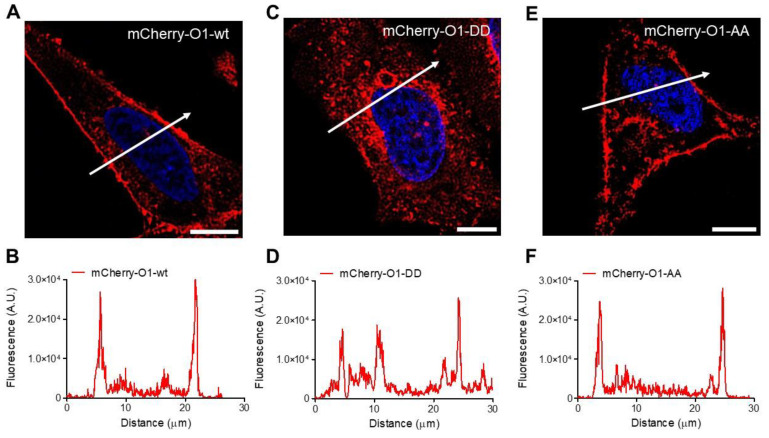
Cellular distribution of the different O1 channels in HeLa cells. Representative fluorescence micrographs for cells in resting conditions expressing mCherry-O1-wt (**A**), mCherry-O1-DD mutant (**C**), or mCherry-O1-AA mutants (**E**). White bars represent 10 μm. White arrows indicate the cell region used to obtain the fluorescence amplitude profile for mCherry-O1-wt (**B**), mCherry-O1-DD (**D**), or mCherry-O1-AA (**F**). Although all Orai channels displayed a certain level of internalization, the internal distribution was the largest in mCherry-O1-DD.

**Figure 6 cells-11-02037-f006:**
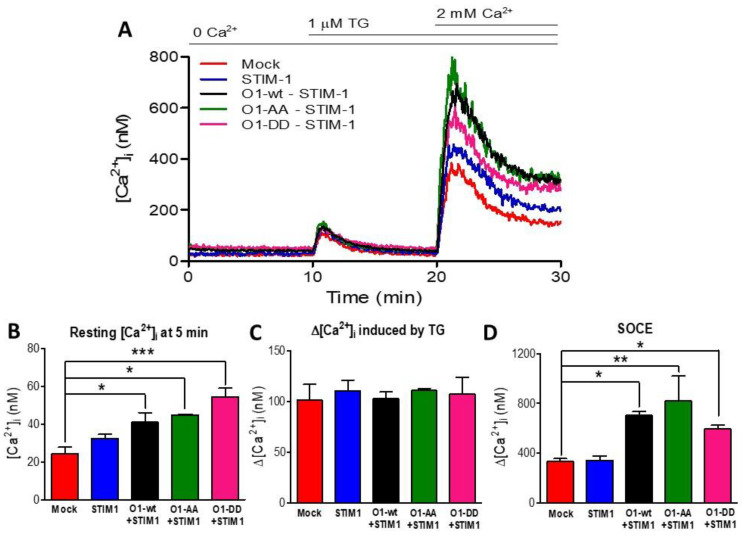
Mutations at residues S27 and S30 of the Orai1 channel did not modify the SOCE response. (**A**) SOCE was assessed in mock cells (red trace) and those transfected with STIM1 (blue trace) or co-transfected with STIM1 and O1-wt (black trace), STIM1 and O1-AA (green trace), or STIM1 and O1-DD (pink trace). (**B**) Despite the absence of external Ca^2+^, the resting [Ca^2+^]_i_ (measured at 5 min) was higher in those cells expressing STIM1 and Orai1 channels but was maximal for those with STIM1 and O1-DD (pink bar). (**C**) The average thapsigargin-induced peak [Ca^2+^]_i_ response was not different among the cells expressing different O1 channels. (**D**) The addition of 2 mM [Ca^2+^] revealed SOCE. The maximal amplitude of the response was obtained by fitting the rate of [Ca^2+^]_i_ increase to a monoexponential equation to calculate the ∆[Ca^2+^]_i_. The combination of STIM1 with any Orai1 channels resulted in the same SOCE response. Bars are the mean ± SEM. * *p* < 0.05; ** *p* < 0.01; *** *p* < 0.001 from at least three independent experiments.

**Figure 7 cells-11-02037-f007:**
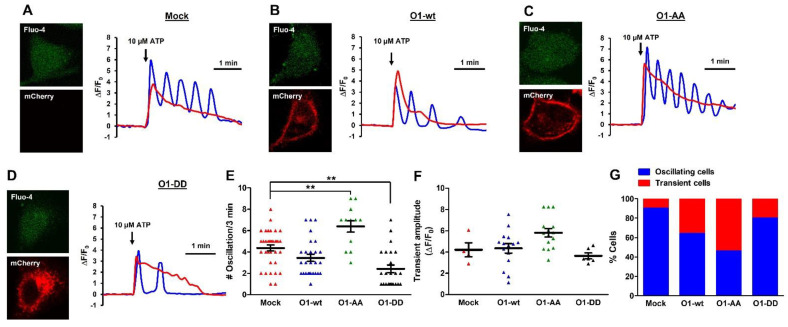
The O1-DD channel inhibited while the O1-AA channel stimulated the frequency of ATP-induced [Ca^2+^]_i_ oscillations in the absence of thapsigargin. ATP (10 μM)-induced Ca^2+^ mobilization displayed two different types of responses, oscillations (blue traces) and a transient elevation (red trace) in mock HeLa cells (**A**) or cells expressing mCherry-O1-wt (**B**), mCherry-O1-AA (**C**), or mCherry-O1-DD (**D**,**E**). To determine whether the [Ca^2+^]_i_ oscillations frequency was modified by the different ion channels, the number of oscillations was determined in the same 3 min period. Mock cells had 4.4 oscillations in this time, while the expression of O1-wt did not significantly modify this frequency (3.4), but O1-AA significantly increased it to 6.4 and O1-DD decreased it to 2.4. The O1-DD channel reduced the frequency to the extent that, now, most cells displayed a single oscillation in the same period. (**F**) The peak amplitude of the transient [Ca^2+^]_i_ response was not significantly modified by expressing any of the O1 channels. (**G**) Percentage of cells either oscillating (blue bar) or with transient [Ca^2+^]_i_ responses (red bar). Traces show representative recordings for each condition. ** *p* < 0.01 for a minimum of four independent experiments.

## Data Availability

Not applicable.

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
