# Peer review of "PKC-Mediated Orai1 Channel Phosphorylation Modulates Ca2+ Signaling in HeLa Cells"

_cells, 2022, doi:10.3390/cells11132037_

Round 1

Reviewer 1 Report

In manuscript by Ericka MatÍnez-MartÍnez et al. they used a combination of agonist induced (ATP) and SERCA inhibitor (Tg) to obtain a rapid and complete depletion of ER calcium store.   Using this protocol, they demonstrated that PKC phosphorylation of ORAI1α stimulates Ca2+ entry and reduces IP3R-mediated calcium release. These findings are not remarkably novel, but the study is well constructed. With mostly minor corrections, I recommend this manuscript for publication in Cells.

Major point:

  • P3 line 4-6/supplementary Figure 1A: The author are only showing representative traces. To claim that overexpression of O1 did not modify the peak, proper peak measurement of 3 independent experiments and statistical analysis should be performed.

Minor points:

  1. Abstract: Define E106Q mutant ( eg. pore dead)
  2. STIM1 abbreviation on page 1 is introduced at line 41, but first used at line 34
  3. Page 2, line 22: change “protein STIM” for “STIM protein”
  4. Supplementary Figure 1: Add Ca2+ concentration in figure legend
  5. Supplementary Figure 1C: Need better explanation of what is measured here. In Method perhaps change the wording Luminal for endoplasmic reticulum lumen. On page 3 line 11, explain what is Mag-Fluo4
  6. Supplementary Figure 1D: add the legend on the figure
  7. Supplementary Figure 2A: Please indicates how many independent experiments were used for analysis and what are the stats applied to this bar graph.
  8. All figure legends: fix formatting issue with ATP and TG concentration, SEM. “Ca2+” should be Ca2+
  9. Figure 1A: add legend to the graph
  10. P3 line 10 and 1: Move Fig1A reference after : “initial rate of calcium rise”
  11. Fig 2A: add legend to the graph
  12. P5, line 7: “we decided to study whether this process is involved in the effect of O1-wt”. The author should complete this sentence to state what is the O1-WT effect they are assessing.
  13. Figure 6a: add the legend on the figure
  14. Figure 6a: change Tiempo (min) for Time (min)
  15. Figure 6b,c,d: please indicate what is measure here on the graph. Eg. add Ca2+ (nM) at 5 mins.
  16. Figure 7. Is red and blue traces an average of all cells or a representative cell? Please add a panel on this figure showing number or % of oscillating cells.

Reviewer 2 Report

Authors have been focused in this article the role of phosphorylation of Orai1 S27/S30 residues in the modulation of Ca2+ signaling in HeLa cells.

I found the manuscript very interesting. The central role of Ca2+ signalling is well known in the regulation of numerous biological processes. However, limited information is available about the regulatory role of Orai1 in the IP3R-mediated Ca2+ release and the molecular background of complex formation. Therefore, the topic is actual and relevant and the results from this study contribute to understand the exact mechanism of Ca2+ signaling.

comments:

  1. The authors showed on supplementary figure 1, that the overexpression of O1-wt in HeLa cells did not modify the peak [Ca2+]i response, but the plato phase show some difference. Did you evaluate the plato phase of the signal?
  2. Did the authors confirm the effeciency of the transfection? If yes, how?
  3. In some experiments the authors used 1.8 mM external Ca2+ contained solution, but in some cases 2 mM. What is the reason of this different external Ca2+ concentration?
  4. The authors demonstrated that phosphorylation by PKC of Orai1 Ser27 and Ser30 residues stimulates the association with IP3Rs. It should be useful to investigate the role of this phosphorylation on other components of this complex as well.
  5. Proximity ligation assay (PLA) was used to study the interaction between Orai1 and IP3 There are other methods to visualize this interaction at even higher resolution, for example dSTORM.
  6. The visible DAPI staining would be informative on Figure 5.

Author Response

Please, see the attached file

Round 2

Reviewer 1 Report

No further comments

Reviewer 2 Report

Thank you for your answers. I don't have any comments.